# Skepticism in the Early Stage of the Introduction of Environmental Enrichment in Japanese Zoos

**DOI:** 10.3390/ani14020309

**Published:** 2024-01-19

**Authors:** Kazuhiko Ota, Saika Yamazaki

**Affiliations:** 1Department of Policy Studies, Nagoya Campus, Nanzan University, Nagoya 466-0824, Japan; 2Inokashira Park Zoo, Tokyo 180-0005, Japan; saika@home.email.ne.jp

**Keywords:** animal welfare, environmental enrichment, Japanese zookeepers, misinterpreting, cultural and historical contexts

## Abstract

**Simple Summary:**

This research explores the initial skepticism and bewilderment among zoo staff about animal welfare and environmental enrichment introduced in Japanese zoos in the mid-1990s. Based on the results of a 2001 questionnaire survey conducted by the Japanese Association of Zoos and Aquariums and publicly available literature, this paper demonstrates that this skepticism and confusion are not simply due to wariness of a new initiative but to a complex combination of factors, including inadequate information, historical and cultural context, and differing perceptions of animal care in the legal system. For example, it was shown that in the early 2000s, environmental enrichment was misinterpreted as exhibit strategy, leading to resistance from zoo staff. This misinterpretation and resistance can be traced back to the post-World War II popularity of mobile zoos and the legal perception of animals as mere “living beings”, not “sentient beings” under Japan’s Animal Protection Law. Through the analysis of a case study in Japan in the early 2000s, this paper aims to identify some of the early challenges to introducing and practicing animal welfare and environmental enrichment in non-Western cultures and contribute to better understanding and practice. Despite regional differences, this study aims to contribute to understanding and improving the universal acceptance and application of animal welfare.

**Abstract:**

This study examines the Japanese zoo staff’s initial skepticism and bewilderment regarding animal welfare and environmental enrichment in the mid-1990s. Utilizing a 2001 questionnaire conducted by the Japanese Association of Zoos and Aquariums and existing literature reveals that this resistance stemmed from inadequate information, cultural history, and legal perspectives on animal care. Specifically, environmental enrichment was initially misunderstood as an exhibition strategy, partly due to post-WWII trends in mobile zoos and legal views of animals as non-sentient under Japan’s Animal Protection Law. The research highlights the early hurdles in adopting animal welfare and environmental enrichment in non-Western settings, aiming to provide insights for other regions dealing with similar transitional challenges. It also addresses misperceptions about environmental enrichment in the context of empathetic relationships and professional ethics in Japanese zoology, offering insights into regions facing similar issues. Additionally, the paper discusses the progression of animal welfare practices in Japanese zoos and related managerial challenges, acknowledging cultural and institutional factors. Despite regional differences, this study aims to contribute to understanding and improving the universal acceptance and application of animal welfare.

## 1. Introduction

The role of zoos and aquariums in supporting species and society is becoming increasingly vital in light of the escalating climate crisis and biodiversity loss [1,2,3]. These institutions are often likened to a modern-day “Noah’s Ark”, playing a crucial role in safeguarding endangered species through strategic conservation and breeding programs [4]. They also serve as powerful platforms for education, heightening awareness about wildlife among the general public and policymakers, especially concerning the climate crisis and biodiversity decline [5]. However, the ethical dimensions of zoos and aquariums are intricate and multifaceted, sparking debates [6,7,8,9,10,11]. There are concerns about captive environments and how they differ from natural habitats could compromise animal welfare. Some establishments may even limit animals’ natural behaviors, using them primarily for commercial uses or entertainment [9,12]. As a result, contemporary zoos must seek a balance: addressing ethical considerations related to animal care and remaining at the nexus of conservation and societal engagements [8,13,14].

One proposed solution is the introduction of environmental enrichment programs, which can enhance visitor experiences and captive animal welfare [15,16,17,18]. Environmental enrichment refers to providing stimuli that promote species-appropriate behaviors and mental activities to bolster animal welfare. By focusing on captive animals’ physical and psychological needs, enrichment aligns directly with ethical considerations, striving to prevent stress, abnormal behaviors, and health problems due to insufficient environmental stimuli.

Such initiatives typically encompass enclosure modifications to promote natural behaviors, varied feeding methodologies, and group housing. Previous research has extensively addressed the design and evaluation of these enrichment strategies due to their profound benefits—such as mitigating boredom and stress and promoting overall health. Today, animal welfare and environmental enrichment are increasingly seen as quantifiable through scientific means, prompting more comprehensive assessment methods [19,20,21].

On the other hand, acceptance and implementation of animal welfare principles, especially in non-Western countries, depend heavily on cultural, economic, and organizational backgrounds. In the early 2000s, studies from Thailand [22,23,24], the Philippines [25], and Korea [26,27] suggested that a region-specific approach is needed to promote animal welfare, and international standards should not be applied but should be tailored to local cultures and circumstances. The Thai case study suggests that while Thailand’s major zoos meet animal welfare assessments based on international standards, these may not be adaptable to all zoos in Thailand and that differences in cultural practices and available resources may present obstacles to achieving animal welfare standards. In the major zoos in the Philippines, while many cages met the minimum standards, local assessors tended to overestimate their zoos. The challenges of standardizing animal welfare and environmental enrichment, particularly the subjectivity of raters and local-specific management practices affecting the application of animal welfare standards, are also relevant to the Japanese case discussed below. The Korean case study confirms that the level of animal welfare varies widely from zoo to zoo and that standards are inadequate in some cases. This suggests standardized standards and organized policies to improve animal welfare. These examples demonstrate that economic constraints, available resources, and perceptions of animals (e.g., primates) within a region influence animal welfare practices and that animal welfare standards and practices require various approaches that vary from country to country and region to region.

Today, global discussions on animal welfare and the intricate nature of captive animal care have urged zoos and aquariums worldwide to recalibrate their practices to align with ethical and scientific standards. However, this transition has been inconsistent across regions, and a gap exists in the literature. Therefore, the experience of Japanese zoos in the 1990s and early 2000s, as they grappled with introducing the concept of environmental enrichment and broader animal welfare discourses, offers an illuminating case study. This paper delves into the challenges Japanese zoos encountered during the early phases of adopting environmental enrichment, drawing from a questionnaire survey in 2001. By examining these initial hurdles, we highlight the complexities stemming from cultural values, historical practices, and emerging global standards. Furthermore, we dissect the misinterpretation surrounding environmental enrichment from three angles: the misinterpretation of environmental enrichment, the empathetic relationships with captive animals ingrained in Japanese animal husbandry, and the professional ethics prioritizing captive animals’ longevity. Through these lenses, this paper aims to offer insights beneficial for other regions navigating similar transitional challenges.

## 2. Stakeholders’ Bewilderment When Environmental Enrichment Was Introduced in Japanese Zoos in the 2000s

### 2.1. Awareness of Environmental Enrichment and Barriers to Implementation

In Japanese zoos, the need to focus on the psychological and spiritual aspects of animals in zoo keeping and to improve their rearing environment has already been pointed out since the 1970s, and attempts have been made to this end [28]. In the 1990s, environmental enrichment was introduced as a movement that took place in American zoos in the 1980s [29]. The commencement of this discourse is often attributed to an influential article published by Matsuzawa in 1996 titled “Animal Welfare and Environmental Enrichment” in the journal of the Tokyo Zoological Gardens Association [30]. Several other important papers have been published, raising awareness in Japan and especially of environmental enrichment [31,32,33,34]. These papers demonstrate the importance of zoos promoting the natural behavior of captive animals using three-dimensional space and changes in feeding methods and emphasize that the introduction of various environmental enrichment techniques will enable Japanese zoos to promote the mental wellbeing and behavioral diversification of captive animals. Following this, in 2001, the non-profit organization “Citizen ZOO Network” was established to advocate for the expansion of these practices, recognizing and sharing the efforts of zookeepers and institutions nationwide that were working on enhancing environmental enrichment.

Nevertheless, during this formative period, a degree of skepticism and resistance was observed within the zookeeper community. This is because they were uncomfortable with the theoretical direction of having animals kept in enclosures that mimic a “natural” lifestyle and trying to match their behavior to patterns observed in their natural habitat [35,36,37]. For example, Hori, a zookeeper at the Tokyo Metropolitan Government, noted that there was an underlying tension amongst his colleagues at that time who felt their longstanding practices were being questioned. When environmental enrichment was introduced, the zoo staff members at work were proud to say, “even if no one tells us to do so, we have always taken care of the animals in our care as a matter of course”.

To provide a more nuanced understanding of the prevalent attitudes during this period, this paper analyzes a 2001 questionnaire survey conducted by the Japanese Association of Zoos and Aquariums (JAZA) [38]. This survey, which forms part of an annual nationwide study on husbandry techniques by JAZA, specifically focused on the implementation of environmental enrichment across 152 member institutions, and the respondents to this survey were employees of these facilities. Despite the low response rate and the survey’s subsequent non-publication, there is limited availability of quantitative data on this topic in Japan, making the findings particularly valuable for understanding the state of environmental enrichment practices during this period.

The survey instrument itself was a structured questionnaire, which incorporated a mix of multiple-choice and open-ended questions. This format allowed for a comprehensive understanding of both the situation of commitment about environmental enrichment and the specific ways in which these practices were being implemented. In compliance with research ethics, all collected data were anonymized to ensure the privacy and confidentiality of the respondents. The survey did not include any personally identifiable information and focused solely on the institutional practices and individual perceptions related to environmental enrichment.

As a result, responses were obtained from 1243 staff members at 88 of the 153 facilities belonging to the association during the survey period. The demographics of the respondents were 20.5% female and 79.3% male. By age group, most staff members were in their 20s and 30s, with 33.3% in their 20s and 30.4% in their 30s. In addition, 84.4% of the respondents were in charge of breeding, and in order of experience, 29.9% had less than five years of experience, 23.7% had more than five years and up to ten years, and 17.3% had more than ten years and up to fifteen years. From the above, the primary respondents to this survey were young and mid-career breeding managers involved in daily breeding operations.

The question about the term “environmental enrichment” and what it means was answered by 1220 respondents. Forty-three percent of all respondents answered “Never heard of it”, twenty-two percent of all respondents answered “Have heard of it but do not know what it means”, and thirty-five percent of all respondents answered “Know what it means”. This suggests that in 2001, when the survey was conducted, awareness of environmental enrichment was considerably lower than it is today [Figure 1].

When staff (n = 694) who answered that they did not implement environmental enrichment were asked why they did not implement environmental enrichment, the most common reasons given were “Lack of budget/equipment” (20%), “Lack of support from managers/colleagues” (19%), “Lack of staff with sufficient knowledge and training” (16%), and “Insufficient manpower” (13%), in that order. The “Other” category includes responses such as “Considering implementation”, “Doubtful that positive effects will be achieved”, and “No particular reason”. (It is unclear whether the phrase “considering implementation” implies that there is still a barrier or that the barrier does not exist and it is just not being implemented yet (Figure 2)).

### 2.2. Reported Cases of Environmental Enrichment Implementation

In the open-ended questionnaire, in response to the question regarding the details of environmental enrichment implementation, a total of 448 cases were reported: 1 arthropod, 1 fish, 5 amphibians, 8 insects, 85 birds, and 348 mammals.

In response to a question regarding the implementation of environmental enrichment, most respondents mentioned installing structures in the release area to provide shade and shelter or introducing foraging devices. Some respondents show that they had appropriate knowledge and skills in implementing environmental enrichment practices. On the other hand, some responses did not necessarily have a direct relationship with environmental enrichment and improvement of the welfare of the animals in their care, such as “The entire park is conducting environmentally conscious exhibits” and “Fences have been removed to make it easier for visitors to touch the animals”. These are not strictly environmental enrichment. Environmentally friendly exhibits may also not include environmental enrichment practices. Furthermore, in all cases, most of the post-implementation evaluations were descriptions and analyses centered on the subjective viewpoints of the implementers, and objective evaluations based on quantitative data were very limited. For example, in the report on the rearing of Canadian geese, a safer environment for the geese is created by using monofilament fishing lines to deter potential predators, such as crows. Additionally, the geese’s habitat was designed to be open, providing a diverse range of stimuli. However, for management purposes, certain interventions, like feather removal, were applied to limit their flying capabilities. Without objective data, it would be impossible to assess the overall impact of these varied measures on the welfare and health of Canadian geese [38].

From these responses, it may be interpreted that, as of 2001, among the zoo staff who did implement environmental enrichment, there were variations in their understanding and application of these practices. It was uncommon for practitioners to have a comprehensive grasp of the quantitative and qualitative impacts of the environmental enrichment programs they administered. This situation indicates that, despite their efforts, at that time, practitioners were still in the stages of effectively applying environmental enrichment techniques for the animals under their care. In addition, a notable characteristic of this survey was its low response rate. Out of the surveys conducted by the Association of Zoos and Aquariums in the three years both preceding and following 2001, six were published in zoo and aquarium journals. The average response rate for these six surveys stood at 92.7%, in stark contrast to the 57.5% response rate for the environmental enrichment survey. Ishida, who participated in administering the questionnaires as a zoo employee at the time, described the reception to the environmental enrichment survey as “indifference”. This response suggests a prevalent skepticism or apathy towards the concept of environmental enrichment during that period [38] Finally, given the 1243 survey respondents, the authors recognize that a more rigorous analysis of variance is needed. For example, the following analysis would provide more insight into the introduction and implementation of animal welfare in different culture areas: the survey on environmental enrichment had a lower response rate than other surveys, but which demographics were more or less likely to respond? Were there differences in responses between men and women? Were there significant differences in the number of people who knew about environmental enrichment when comparing early-career staff to more experienced staff? What characteristics led staff to try to practice enrichment? Which characteristics made them perceive the lack of support as a barrier? (These are suggestions from the reviewers. The authors appreciate this feedback). However, as mentioned earlier, the dataset used in this paper results from a survey conducted by JAZA in 2001, and the authors cannot access the complete dataset. The authors plan to conduct an extensive survey and analysis of variance in a future study to analyze the current situation and identify changes over the past 20 years. In addition, the authors are unable to ascertain what ethical approval process it went through. However, given that it is part of a regular survey conducted with the approval of JAZA, that the responses were made with the voluntary consent of the participants, and that the information collected in the survey was processed in a manner that does not identify individuals, we are certain that the ethical validity of the study meets current ethical standards.

## 3. Misinterpreting Environmental Enrichment: Historical Background of Acceptance in Japanese Zoos

### 3.1. Lack of Accurate Information on Animal Welfare Principles: “Environmental Enrichment as a Mere Exhibition Method?”

When the practice and theory of environmental enrichment were introduced to Japan in the 1990s, as noted in the previous section, they elicited diverse reactions among zookeeping staff. This section delves into the factors and historical background behind some of these responses, particularly the negative ones.

Throughout the 2000s, environmental enrichment became prevalent in Japanese zoos. However, misconceptions persisted about this innovative approach. Despite the growing recognition of “environmental enrichment” among Japanese zookeepers and its increasing implementation in daily operations, skepticism was prominent. For instance, a 2008 lecture on evaluation methods for environmental enrichment at the Society for Breeding Technology Conference faced criticism. The lecture featured researchers engaged in monitoring reproductive physiology and conducting assessments using behavioral indicators and husbandry staff who apply environmental enrichment in practice. During the discussion session involving audience participation, skepticism emerged from the attending husbandry staff. One prominent concern raised was the potential bias in behavioral observations used for evaluation. Participants pointed out that these observations might inadvertently focus on behaviors that are conveniently interpretable by humans, thus questioning the objectivity of the assessments. This criticism highlighted a lack of trust in the methodologies of environmental enrichment among practitioners [38].

Moreover, during a 2009 lecture in Japan by the late Valerie Hare, the Executive Director of the American non-profit organization Shape of Enrichment, diverse interpretations of environmental enrichment among zookeepers emerged. During a pre-lecture workshop, some of the participating Japanese zoo staff members commented on the effectiveness of environmental enrichment in enhancing the exhibition of animals’ diverse behaviors for visitors. This indicated a possible conflation of environmental enrichment with performance-oriented display techniques in zoos. Furthermore, after Hare’s keynote speech on the state of environmental enrichment in the United States, a question-and-answer session ensued. A significant query raised by a managerial staff member involved the extent of human intervention in wild animal care, especially in cases like target training for polar bears at San Diego Zoo, and the ethical boundaries of such involvement, particularly for animals with no prospect of returning to the wild. Hare discussed the practice of habituation training for exhibit animals that interact with visitors, suggesting that pre-exposure to human contact can mitigate welfare issues at the exhibition stage, and emphasized the need for appropriate training based on the intended use of the animals. This interaction highlighted a fundamental discrepancy in perspectives between the Japanese zoo staff, questioning the degree of human intervention in wildlife care, and Professor Hare, who advocated for management practices tailored to the animals’ nature [38].

These varied interpretations can be attributed to misinterpretations and cultural apprehensions surrounding animal welfare and environmental enrichment. A significant factor was the introduction of environmental enrichment when accurate information on animal welfare was scarce. Consequently, environmental enrichment, fundamentally a husbandry method, was often mistaken as a method for showcasing animal behavior. Such misinterpretations can be traced to the historical context of Japanese zoos, which were traditionally seen more as entertainment venues than research and conservation sites. For example, the popularity of circus-like mobile zoos in the 1950s cultivated an expectation among visitors for show-like exhibitions. In the 1990s, Japanese zoos received criticism from animal rights groups and began reducing the number of entertainment-focused exhibits. As environmental enrichment was introduced, they were sometimes misconstrued as a continuation of the “animal show” paradigm.

This misinterpretation stems from a lack of clarity about the animal welfare philosophy underlying environmental enrichment. This philosophy has informed the practice since the 1990s, following discussions and studies on animal welfare from the 1960s onward [39]. In Japan, however, environmental enrichment was introduced without the foundational discussions on animal welfare. As a result, it was frequently misinterpreted as a mere exhibition technique [37]. Moreover, it was often confused with “landscape immersion” [40], an exhibition method introduced to Japanese zoos in the 1990s, leading to further confusion among staff. Additionally, as mentioned above, as Japanese zoos were transitioning away from entertainment-centric exhibits in the 1990s in response to animal welfare concerns, the misinterpretation of environmental enrichment as a mere exhibition method triggered skepticism among staff in the 2000s.

To exemplify, in 1999, Ueno Zoological Park introduced a bamboo tube feeder as an environmental enrichment trial for giant pandas based on a recommendation from the San Diego Zoological Society. While this increased the pandas’ foraging time and made their behavior more akin to wild pandas, the zookeeper who introduced the program later expressed worries that visitors might perceive the pandas’ behavior as a mere spectacle [41]. While environmental enrichment’s primary goal is fostering natural animal behavior by enhancing their captive environment, this approach was misread by zoo staff as a choreographed display, leading to concerns about violating the animals’ dignity.

### 3.2. Empathetic Relationships in Animal Husbandry: Differences between Sewa and Environmental Enrichment

To understand the nuances of introducing environmental enrichment in Japanese zoos, it is essential to delve into the historical background of Japanese attitudes toward captive animals.

Japan has traditionally not been a meat-eating nation and lacks a production system where animals are managed in the field for subsistence, as is common in the West [42]. For instance, native cattle in Japan were primarily raised for labor during the Edo period (1603–1868), serving as service animals and a source of fertilizer. These cattle were tamed like members of the family [43]. Folk songs from that era, such as “Nanbu Ushioi-Uta”, sung by cattle drovers transporting goods, convey concern for cattle’s welfare: “Our journey is hard. The long journey of seven days and seven nights. It must be hard for you, cattle. Patience for the moment” [44]. Japan’s contemporary harvest and fertility festivals continue gratitude to these draft cattle. These customs and songs reflect a sympathetic relationship between humans and animals in Japanese agricultural production. For example, the “Ushigoe Matsuri” held at Sugawara Shrine in Miyazaki Prefecture is a festival to pray for the health of livestock and a good harvest; it has a history of more than 400 years, and at the end of the Edo period, about 600 decorated cattle participated in the festival.

Various other expressions of empathy for farmed animals are found in traditional Japanese perspectives. For example, many Japanese regions still observe a horse memorial service. Other examples of animal memorials are the animal memorials found in zoos, research institutes, and other animal breeding facilities in Japan. These memorials are primarily intended to show appreciation for animals sacrificed during breeding or animal experimentation and to comfort their souls [45,46,47]. Moreover, the Hōjō-e ceremony emanates from the Buddhist precept against taking life. This ceremony involves releasing some the captive animals, usually birds or fish, back into the wild, fostering compassion and merit for the individual releasing the animal and the broader community. Historically, this also served as atonement for the sin of killing animals.

Japan’s sympathetic approach towards captive animals aligns well with emotional animal protection. The Western-style animal protection movement began in Japan in the 1890s. Spearheaded by foreign students and religious leaders, the Society for the Prevention of Cruelty to Animals (later renamed) emerged, heavily influenced by Christianity and supported by the middle and upper classes. The contemporary “Law Concerning the Protection and Control of Animals” was formulated under this movement’s influence [48].

In modern Japanese society, this empathetic attitude persists, although it is evolving within the livestock industry [49]. In zoos, many Japanese zookeepers affectionately refer to the animals in their care by a nickname and do “sewa” [50]. While “sewa” translates to “care”, it carries a more profound connotation, like a familial affinity or kinship [50]. In contrast, the term “husbandry” is traditionally used in contexts to describe systematic behavior management techniques, such as health management (including blood sampling) and reproductive management (like sperm collection) [38]. Thus, by the late 1990s, Japanese zookeeping practices balanced the establishment of empathic relationships with animals and a clear distinction between husbandry techniques intended for explicit animal management.

However, environmental enrichment does not neatly fit into either the “sewa” or “husbandry” categories, possibly leading to confusion about its role [38]. This suggests that the empathetic relationship with captive animals was different from the objective concept of environmental enrichment or specific care methods.

### 3.3. Professional Ethics Emphasizing Longevity: “Living Beings”? “Sentient Beings”?

Other points worth noting are that the theoretical framework encompassing the concepts of animal welfare and environmental enrichment did not always align with the predominant view among Japanese zookeepers, who prioritize ensuring captive animals live for an extended period [38,51].

This stance, which gravitates away from euthanasia and emphasizes the longevity of captive animals, is mirrored in Japan’s animal management legislation. The Act on Welfare and Management of Animals, enacted in 1973 and revised in 1999, applies to domestic, display, industrial (livestock), laboratory, and other animals under human care. It defines these animals as “living beings” [52]. This characterization contrasts with the Amsterdam Convention, which was established in 1999 and became the EU’s foundational principle for animal welfare. The convention recognizes domestic animals as “sentient beings”, warranting the utmost consideration by influencing policy changes concerning domestic animals across numerous countries. Thus, in Japan, the emphasis lies more on animals being “living beings” rather than “sentient beings”, with the primary responsibility of caregivers being to extend their lives.

However, this perspective is changing. A partial revision to Japan’s Act on Welfare and Management of Animals in 2019 incorporated more aspects of animal welfare. This update introduced several modifications, such as revised standards for the proper care of animals, prohibitions on the care and keeping of specific animal species, and clear guidelines on the care and management of dogs and cats. These changes reflect a growing emphasis on longevity and the quality of life of the animals [53].

## 4. Environmental Enrichment and Animal Welfare in Japanese Zoos: Management Issues for Realization

In previous sections, we highlighted the initial resistance to environmental enrichment in Japanese zoos during the 2000s and its underlying reasons. Since then, significant shifts in perceptions and practices have been observed.

The Japan Association of Zoos and Aquariums (JAZA) is proactively working on establishing welfare guidelines in sync with global standards, particularly those of the World Association of Zoos and Aquariums (WAZA). Since 2016, JAZA has organized conferences and workshops on animal welfare. It aims to complete assessments of all member institutions by 2023, and, to this end, it has updated its Code of Ethics and Welfare, formulated animal welfare standards, and drafted guidelines for breeding facilities and conditions suitable for about 90 species. Moreover, it is studying euthanasia standards and revising membership assessment criteria to ensure better animal welfare over the next decade.

Furthermore, the “Law Concerning the Protection and Management of Animals” underwent amendments in 2019 to become more animal welfare-centric, with regional municipalities following suit. For instance, the city of Sapporo in Hokkaido pioneered a new zoo ordinance in April 2022 emphasizing animal welfare. This ordinance, initiated due to past zoo incidents and rising welfare concerns, applies to all municipal or private facilities and covers zoos, aquariums, and insectariums. It mandates that these institutions have clear objectives to ensure animals do not experience pain or distress and can behave naturally. The act discourages anthropomorphism and unnecessary human–animal contact and emphasizes biodiversity conservation. Additionally, there is an emphasis on exhibits reflecting animals’ natural ecology and employing knowledgeable staff for captive animal care. This new ordinance highlights the importance of animal welfare and delineates the roles, philosophy, and future direction of zoos. It underscores the societal expectation that zoos and aquariums should be more than mere viewing platforms; they should educate visitors about biological diversity and ecology [54].

Significant challenges remain in Japanese zoos’ management structure and funding methods, intricately linked with the renewal of facilities. For instance, Sadotomo and Ishida highlight that many public zoos rely heavily on local government budgets. This reliance often prevents these zoos from making ample investments in facility renewal and animal welfare enhancements. This situation starkly contrasts with the well-established management models in the US and Germany. In these countries, NPOs dedicated to objectives like biodiversity conservation have taken the reins of zoo operations. Their financial pillars are admission fees, local government subsidies, and citizen donations [55].

Ishida emphasizes that Japanese zoos seldom operate independently. Most public zoos depend on local governments for their budgets. Meanwhile, private zoos typically operate under the aegis of their parent companies, constraining their access to needed funding. This challenge stems from the historical pricing structure of Japanese zoos. Until the 1950s, these zoos were self-financed. However, during rapid economic expansion, admission fees were significantly reduced, like free entry for children and the elderly. This income model led to delays in facility upgrades and a reliance on aging infrastructure. Consequently, animal welfare and environmental enrichment have suffered.

Additionally, since around 2000, Japanese zoos have increasingly outsourced tasks and turned to non-regular staff to curb labor costs. This shift towards casual labor is a trend observable in zoos and across various Japanese government departments. As a fresh approach to management, the designated manager system is gaining traction. However, it presents unique challenges in zoo management. Given the periodic re-election of the designated manager, fostering trust and transferring expertise become problematic. Successful zoo breeding initiatives hinge on trust-based relationships, like breeding loans between zoos, and the invaluable experience and skill set of the staff overseeing them. Breeding certain species, such as koalas, demands specialist knowledge and capabilities, like ensuring a consistent eucalyptus supply and offering cultivation guidance to adapt to environmental shifts.

Moreover, the constant rotation of keepers is detrimental because a bond of trust between animals and their keepers is crucial. Sadotomo is concerned that the designated manager system may not be best suited for zoos [56].

In more positive developments, since the 2000s, the volume of peer-reviewed papers from fields directly associated with animal care has seen a marked rise for Japanese zoos. Nevertheless, collaboration between zoos and research institutions is not always seamless. While many zoos in Europe and the US trace their roots back to “zoological societies” with researchers as integral members, Japanese zoos, especially those run by local governments, face challenges in attracting such expertise. Research often gets relegated to a volunteer-driven endeavor, impeding the promotion of research.

When juxtaposed with their European and American counterparts, Japanese zoos are in a “low-cost, low-quality” scenario. This situation has led to delayed facility renewals, an increasing reliance on non-regular staffing, and discontent among citizens, governmental bodies, and zoos. Enhancing environmental enrichment and animal welfare within this framework becomes daunting. Potential fundraising avenues, like Western models, include campaigns and crowdfunding. For example, the National Museum of Nature and Science raised over JPY 700 million through crowdfunding in August 2023. Such case studies indicate the need for a comprehensive review of Japanese zoos’ operational and fundraising strategies.

## 5. Conclusions

### 5.1. Summary: Historical and Cultural Background That Caused Bewilderment When Introducing Environmental Enrichment in Japanese Zoos in the 2000s

Based on a survey regarding the early introduction of environmental enrichment in Japanese zoos, this paper examines the challenges and misinterpretations and their primary causes. When environmental enrichment was first introduced around 2000, more than half of the zoo staff were unfamiliar. When attempts were made to implement it, many needed help to garner the understanding of their colleagues and superiors and secure the necessary budget. Moreover, environmental enrichment has often been misinterpreted. This misinterpretation may have been induced mainly by the following factors: 1. A prevailing lack of accurate information on animal welfare principles leads many to misinterpret environmental enrichment merely as an exhibition method. 2. Confusion surrounding the true nature of environmental enrichment. It is neither “sewa” (empathic relationship in animal care) nor “husbandry” (systematic captive animal management). 3. A professional ethos that prioritized the longevity of captive animals.

The Japanese case study shows potential obstacles when integrating Western animal welfare and environmental enrichment concepts into non-Western settings. In Japan, even though understanding and adopting animal welfare and environmental enrichment have increased since the late 2010s, zoos, funded mainly through taxes and corporate investments, often delay facility upgrades. This deferment adversely affects animal welfare and conservation activities. Previous studies emphasize the need for increased funding for human resource development and facility renovations to enhance animal welfare, potentially through higher admission charges and donations. Furthermore, while increasing, animal welfare research in Japan remains limited due to the lack of researchers to fill the roles of curator and director.

This paper emphasizes the need to recognize and address each region’s unique historical, cultural, and institutional nuances when introducing the animal welfare concept. Our findings indicate that integrating these principles of animal welfare into practice can sometimes be rife with misinterpretations and tension. Hence, it is essential for synergistic efforts involving zoo professionals, ethicists, and policymakers to assess the resources and challenges intrinsic to this integration. Collaborative endeavors of this nature can offer clarity and adaptability, all while upholding the zoo animal welfare guidelines. They will invigorate ongoing dialogues and fortify the indispensable role of animal welfare, ensuring that endeavors in this direction are more rewarding.

### 5.2. Future Discussion

Finally, topics that could not be fully explored in this paper are specified. First, although animal welfare and environmental enrichment practices were not smoothly introduced in Japanese zoos, Japanese animal welfare orientation is influenced by Shintoism, Buddhism, and Confucianism, which offer a unique perspective. This spirit, known as “aigo” in Japanese, has influenced the development of applied animal behavior science in Japan [57,58]. Itoh also shows how Japan’s zoo policies and tragic events during World War II influenced post-war perceptions of animals held in zoos [59]. In 1943, the Japanese government ordered the disposal of “dangerous animals” held in zoos to prevent animals from escaping due to air raids. For example, animals at Tokyo’s Ueno Zoo were classified and killed by poisoning, suffocation, or stabbing. Elephants were killed in cruel ways such as starved to death without food and water. This killing was a deep emotional shock to zoo staff and citizens alike.

Next is an international comparison of animal welfare perspectives to understand the attitudes of Japanese zoo staff. Bacon et al. conducted semi-structured interviews with zoo staff in Europe and China and found different attitudes toward animal welfare between the two [60]. Phillips et al. also surveyed attitudes toward animal welfare and rights at universities in 11 European and Asian countries and found that European and Asian students have different attitudes toward animal welfare [61]. This is most likely due to regional sociopolitical conditions rather than differences due to religion. Concerns about animal welfare and rights were also positively correlated with student spending. Nakajima et al. noted differences between Japanese and American students’ views of animal intelligence, which is important for understanding cultural differences in animal welfare perceptions [62]. In a study of how university students in Japan and the US evaluate animal intelligence, they found that, in general, American students rated animal intelligence higher than Japanese students, and females tended to rate animal intelligence higher than males. The animals considered highly intelligent were nearly identical in both countries, but Japanese students rated crows higher than American students. Crows play an important role in traditional Japanese culture and mythology.

Also meaningful is the relationship between staff demographics and attitudes; Randler et al. assessed attitudes toward animal welfare (AWA) among university students in 22 countries [63]. The results showed that the most significant determinant of AWA was eating habits, followed by country and gender. Regarding eating habits, vegans had the highest AWA, and females had significantly higher AWA than males. Whether gender and eating habits influence attitudes toward animal welfare among Japanese zoo staff is an open research question.

Finally, the future of animal welfare and the development of initiatives and evaluation frameworks for its realization can be mentioned. Rose and Riley argue that this should be integrated into zoo operations, with an emphasis on animal welfare and human wellbeing [64,65]. Modern zoos focus on four main objectives: conservation, education, research, and recreation. However, there is debate about whether zoo activities fully achieve these objectives, and their argument that including wellbeing as an objective can improve both animal welfare and human wellbeing provides insight into how zoos in Japan should respond to these goals (in addition, the terms animal welfare and animal wellbeing are used interchangeably [66]). Brando and Buchanan-Smith point out that to provide good welfare for captive animals, zoos should consider their biological and psychological requirements, accommodate the natural features of their life cycles, and account for day/night and seasonal changes [67]. Their proposed 14-criteria-based tool for assessing animal welfare will help Japanese zoos determine whether they are meeting animal welfare needs. In addition, Mellor et al. outline the 25-year history of the Five Domain Model for assessing animal welfare and then present a model for assessing animal welfare that includes human-animal interactions, which will provide an important framework for assessing Japanese zoos’ practices [19].

Ultimately, the authors believe that conducting these Japanese case studies will help to ensure that animal welfare and environmental enrichment practices have value across cultures.

## Figures and Tables

**Figure 1 animals-14-00309-f001:**
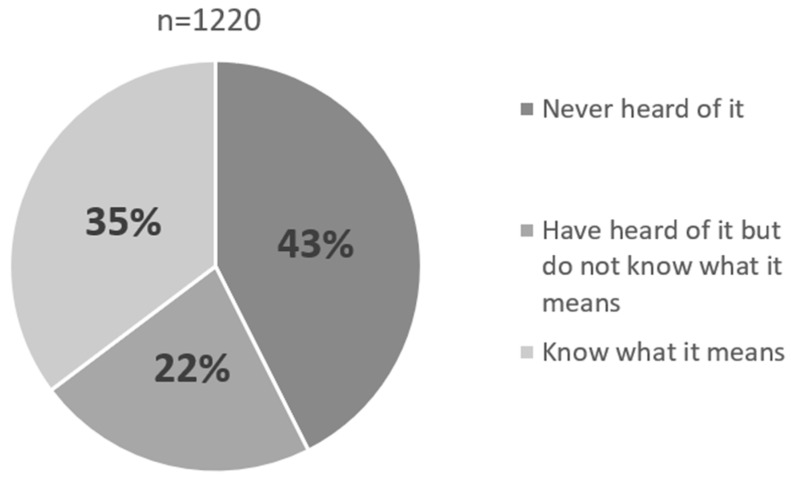
How well recognized was the term “environmental enrichment” in Japanese Zoo Staff, 2001 (Created by the author).

**Figure 2 animals-14-00309-f002:**
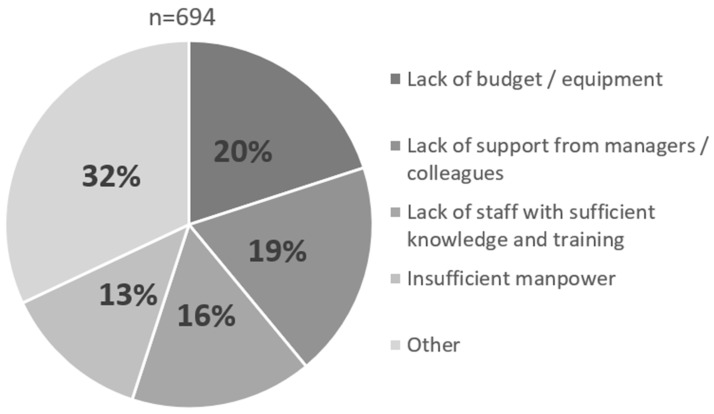
Reasons for not implementing environmental enrichment in Japanese Zoo Staff, 2001 (Created by the author).

## Data Availability

Restrictions apply to the availability of these data. Data were obtained from JAZA and are available from the authors with the permission of JAZA.

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
