# Peer review of "Skepticism in the Early Stage of the Introduction of Environmental Enrichment in Japanese Zoos"

_animals, 2024, doi:10.3390/ani14020309_

Round 1
Reviewer 1 Report
Comments and Suggestions for Authors
This article focuses on the early stages of implementing environmental enrichment in Japanese zoos. The basis of the article is unpublished survey data from 2001, which the authors did not have control of as it was conducted by the Japanese Association of Zoos and Aquariums. As the authors are working with historical data, this limits the control they have over the methods and the questions contained in the survey. However, I still feel this article has important implications for the environmental enrichment space due to the continued growth in understanding of animal welfare that is needed across the world.
For the survey data being a strong component of the abstract/simple summary, it did not feel integrated into the latter half of the article. I don’t know if there is a way to integrate the results more fully throughout the article, or I would suggest restructuring the abstract.
Overall, I think there was good coverage of the literature to my knowledge, the one area that the authors may want to investigate was the 14th International Conference on Environmental Enrichment held in Kyoto, Japan, in 2019.
I have included my more specific comments below.
Simple Summary and Abstract:
Line 9: “Misinterpreting” seems to be used as a noun here and in the abstract (Line 30). I see that you expand on what you mean by “Misinterpreting” in the introduction (Lines 82 to 85). However, as you don’t have space to explain that in the simple summary and abstract, and this is not a common term for most people, I would stick with using “misinterpretation” in these spaces.
Line 30: See above.
2. Stakeholders’ Bewilderment…
Line 111: I think there is an extra word here. More specifically, “because” seems out of place.
Lines 113 to 123: These two paragraphs seem like they could be rewritten into one concise paragraph.
Lines 153 to 154: There seems to be an unintentional repeat in the text here, starting with the second 19%.
Line 174: “tegus” is not a term I am familiar with, and when I researched it, I found more information on the animal tegus as well as an investment program. I suggest including a description of what a tegus is.
Line 172 to 176: I don’t know that I entirely understand the purpose of including the example of the Canadian geese breeding in this article. I think this section could potentially be trimmed down with less details but more emphasis on the purpose of this example.
Line 183 to 184: I believe there is either a translation mistake here or there are some missing words. This last sentence reads to me that it was apparent that each care member was grasping or reaching for animals in their charge.
Lines 179 to 195: These two paragraphs could be streamlined into one as there is some repeating information between the paragraphs.
3. Misinterpreting about Environmental Enrichment
Line 210: It is mentioned that environmental enrichment at the conference “faced criticism”. This needs to be expanded on. What criticism was faced?
Line 212 to 213: Please elaborate on the “diverse interpretations” as well.
Comments on the Quality of English Language
The article has grammatical errors, specifically around the capitalization of certain words that should be addressed before being published.
Author Response
I apologize for the delay in responding to your review. And I sincerely appreciate your careful observations and comments. Based on your comment, I rewrote it as follows.
==
Simple Summary and Abstract:
Line 9, 30: Thank you for pointing out the use of "Misinterpreting" as a noun. It was corrected to "misinterpretation".
2. Stakeholders' Bewilderment:
Line 111: Thank you for pointing out that "because" was misused. This word has been deleted.
Lines 113 to 123: Reorganized these two paragraphs into one concise section.
Lines 153 to 154: Thank you for pointing out the repetition in the text. Duplicate parts have been removed.
Line 174: Sorry for the confusion regarding the term "tegus." This is a "monofilament fishing line". I have corrected the wording.
Lines 172 to 176: Thank you for pointing out the example of breeding Canada geese. I've rewritten this section.
Lines 183 to 184: Thank you for pointing out any translation errors or missing words. I have corrected this sentence.
Lines 179 to 195: Removed duplicate information and combined two paragraphs into one.
3. Misinterpreting about Environmental Enrichment:
Line 210: Specific criticism regarding environmental enrichment was added at the conference.
--
For instance, a 2008 lecture on evaluation methods for environmental enrichment at the Society for Breeding Technology Conference faced criticism. The lecture featured researchers engaged in monitoring reproductive physiology and conducting assessments using behavioral indicators, and husbandry staff who apply environmental enrichment in practice. During the discussion session involving audience participation, skepticism emerged from the attending husbandry staff. One prominent concern raised was the potential bias in behavioral observations used for evaluation. Participants pointed out that these observations might inadvertently focus on behaviors that are conveniently interpretable by humans, thus questioning the objectivity of the assessments. This criticism highlighted a lack of trust in the methodologies of environmental enrichment among practitioners.
--
Line 212 to 213: Also added details about "diverse interpretations."
--
Moreover, during a 2009 lecture in Japan by Professor Valerie Hare, the Executive Director of the American non-profit organization Shape of Enrichment, diverse interpretations of environmental enrichment among zookeepers emerged. During a pre-lecture workshop, one part of the participating Japanese zoo staff members commented on the effectiveness of environmental enrichment in enhancing the exhibition of animals' diverse behaviors for visitors. This indicated a possible conflation of environmental enrichment with performance-oriented display techniques in zoos. Furthermore, after Professor Hare's keynote speech on the state of environmental enrichment in the United States, a question-and-answer session ensued. A significant query raised by a managerial staff member involved the extent of human intervention in wild animal care, especially in cases like target training for polar bears at San Diego Zoo, and the ethical boundaries of such involvement, particularly for animals with no prospect of returning to the wild. Hare discussed the practice of habituation training for exhibit animals that interact with visitors, suggesting that pre-exposure to human contact can mitigate welfare issues at the exhibition stage, and emphasized the need for appropriate training based on the intended use of the animals. This interaction highlighted a fundamental discrepancy in perspectives between the Japanese zoo staff, questioning the degree of human intervention in wildlife care, and Professor Hare, who advocated for management practices tailored to the animals' nature.
--
Reviewer 2 Report
Comments and Suggestions for Authors
The issue raised by the paper is highly relevant, as it highlights cultural specificities that influence the uptake of practices proposed internationally to improve animal welfare (in this case in zoos). The study itself lacks scientific quality. The main problem is that its discussion relies heavily on unpublished data from a survey carried out in 2001. It discusses the results of the survey without showing methodological details. Importantly, there is no comment of the ethical appraisal of the experimental protocol, although it deals with research involving humans. Therefore in my opinion the paper cannot be accepted for publication.
Author Response
I apologize for the delay in responding to your review. And thank you very much for your valuable feedback. In response to your comments, I would like to address the following points:
- Yamazaki and I are grateful for your recognition of the importance of cultural specificities in the international implementation of practices to improve animal welfare in zoos.
- Regarding your point about the heavy reliance on unpublished survey data from 2001 and the lack of methodological details, Yamazaki and I enhanced the credibility of the research by providing additional information on the survey methodology.
- We appreciate your essential point regarding the lack of comments on the ethical evaluation of research involving human subjects. Since this is the result of a study done by JAZA in 2001, we cannot ascertain what ethical approval process it went through. However, given that it is part of a regular survey conducted with the approval of JAZA, that the responses were made with the voluntary consent of the participants, and that the information collected in the study was processed in a manner that does not identify individuals, we believe that the ethical validity of the study meets current ethical standards.
Reviewer 3 Report
Comments and Suggestions for Authors
Thank you for the invitation to review this very interesting paper. Explorations of diverse cultural perspectives on animal wellbeing are vital to connecting the worldwide zoo and aquarium community as we work together to meet the same goals and optimise the care of animals living in our care and are always welcome in the peer-reviewed literature. The zoo and aquarium research field is, often, dominated by those works from European and North American members of the community, and as one zoo global zoo community the diverse cultural perspectives and voices that can inform the ways forward together are integral to ensuring optimal wellbeing of all animals, all over the world.
The presented case study is an interesting one. An examination of the early stages of adopting new animal care practices, in this instance enrichment, provides a useful insight into how attitudes towards animal wellbeing shift and change in the beginnings of new best practices. Broadly, exploring these attitudes and how they are shaped by cultural and societal factors is useful to the wider animal care community; there is often a gap between ‘science’ and ‘practice’ marked by the various barriers described in the present survey. Understanding how these barriers can be overcome in pursuit of the worldwide advancement of animal wellbeing science is integral to adopting more modern, ethical, and compassionate animal care practices. To that end, this paper is at its core suitable for publication, however, there is context missing that would enhance the discussion and lead to more meaningful recommendations and conclusions. For example, Western values on animal welfare are commented on throughout, without significant introduction which highlights exactly what these values are and where they differ from those in Japan. Some useful references are missing from the discussion; some are highlighted in the comments below.
There are some minor grammatical errors throughout; a few of the more obvious ones have been highlighted in the specific comments. Overall, the English language in the manuscript is concise and comprehensible but would benefit from some minor editing to ensure clarity.
General comments
The study methodology is sound and comprehensive. It would be useful to know if the demographics of the survey respondents broadly match up with the demographics of zoo staff at the time, particularly with regards to age, sex, and experience, particularly as the authors note a low response rate in comparison to other surveys of the period. How do this survey’s demographics match up with those? Were a certain group more likely to respond than others, for example?
Statistical analysis is lacking and could be useful to better understand where different responses were coming from. For example, were there differences between male and female responses? Were there significant differences between young, early-career caregivers and more experienced caregivers in how many respondents knew what enrichment was? Were certain demographics of caregivers more likely to try implementing enrichment in practice? Which demographics were more likely to feel that lack of support was a barrier? Answering these types of questions with a more comprehensive analysis would greatly benefit the applicability of the research to improving animal wellbeing.
Overall, the study is interesting and presents some introductory concepts on the topic but is missing a deeper exploration of the subject that would make the manuscript more impactful. The article would benefit from making it clear that enrichment should just be one part of a larger and holistic animal wellbeing program that considers all aspects of the animals’ care. Enrichment is very important but is not and should not be the only solution available for zoos to ensure animals are healthy and happy when living in human care. Attitudes towards enrichment in general perhaps represent a wider trend in how animal wellbeing and animal care practices that centre on the animal’s experience are seen in different cultures. This much is reflected in worldwide attitudes towards anthropomorphism and a historical reluctance to attribute qualities such as the ability to experience emotions and affective states to non-human animals; perhaps such trends are worth discussing here, too.
Itoh Mayumi wrote a comprehensive book on Japanese wartime zoo policy which would aid in providing much-needed context of how the post-WWII era shaped zoo legislation in Japan. As this is mentioned in the abstract, a more thorough discussion of the topic was expected. There are also some useful articles that are not currently discussed that may be relevant when comparing cultural attitudes towards animal wellbeing:
Bacon et al. (2021) discussed the differences in attitudes towards the significance of their work between European and Chinese zoo professionals: https://www.mdpi.com/2673-5636/2/4/46
Nakajima et al. (2015) compared estimates of animal intelligence between students from Japan and the United States: https://doi.org/10.2752/089279302786992504
Randler et al. (2021) looked at animal welfare attitudes (AWA) in students from around the world with a focus on gender as a predictor, but with much useful discussion about cultural differences that may influence AWA. As many of the survey respondents in the present study were male this paper is certainly worth exploring: https://doi.org/10.3390/ani11071893
Phillips et al. (2012): https://www.ufaw.org.uk/downloads/awj-abstracts/v21-1-phillips.pdf
It would also be pertinent to include a more thorough discussion on what, exactly, ideal animal welfare principles look like to bolster the discussion of the paper. The introduction is missing some groundwork for highlighting what optimal animal wellbeing looks like to more fully integrate these ideas into the discussion and conclusions. We know what the past is, and we know where we are right now, but where do we need to go? Do the survey results indicate opportunities and challenges for progressing towards those goals for Japanese zoos in particular? Around the world? What are the broad animal welfare goals that all zoos worldwide should be working to achieve, and how is that accomplished? There are many useful articles that will help contextualise the ‘where we should be going’ discussion that surrounds animal wellbeing and zoos and aquariums in general. Here is a selection of just a few:
Rose and Riley (2022): https://www.frontiersin.org/articles/10.3389/fpsyg.2022.1018722/full
Brando and Buchanan-Smith (2018): https://doi.org/10.1016/j.beproc.2017.09.010
Mellor et al. (2020): https://doi.org/10.1016/j.beproc.2017.09.010
There are certainly many more papers on the topic and these serve as a starting point for the authors to explore the topic in greater detail in the context of the importance of enrichment to animal wellbeing and animal wellbeing in general.
Finally, Sato wrote a wonderful chapter on the role of ‘aigo’ in the development of animal wellbeing science in Japan in the 2016 book “Animals and us”. The concept of aigo has been discussed in some interesting recent papers from Kenichi Shuntō which may also warrant further exploration: https://researchmap.jp/70875319
Line by line comments
Line 9, Line 30 – ‘Misinterpreting’ does not need to be capitalised.
Line 44 – 45 – Personally I would combine these into one paragraph. Some of the paragraphs in the introduction are quite small and they connect quite well, so could easily be merged.
Line 52 – If both animal welfare and animal wellbeing are going to be used throughout this paper I would clarify this by stating this clearly and for example such Moberg and Mench 2000 as a reference.
Line 58 – 59 – See above comment.
Line 66 – 67 – This sentence is unclear; perhaps something like “The acceptance of animal welfare principles, especially outside the West, is highly variable dependent on cultural context” would open the paragraph better. The paper would also benefit from a
Line 67 – 68 – What exactly did these studies find? What reasoning is provided for the results of these studies? Are there any similar studies of viewpoints from Western zoo communities?
Line 69 – What Western cultural and historical frameworks, exactly? Can you provide some examples – maybe a brief history would be useful?
Line 70 -71 – This is an interesting point and could really do with being expanded upon. What are the key ways that the discussed cultures differ from those in the West? Points could be made here about differences in spirituality, collectivist vs. individualist cultural viewpoints, political climates… What values dissonances exactly lead to differences in attitudes towards animal welfare?
Line 74 – Perhaps the 3Rs are worth discussing here in relation to scientific standards, as these can be extended to zoos and aquariums; https://link.springer.com/article/10.1007/s10806-022-09892-5
Line 99 - 101 – More detail would be useful; how exactly did these papers shape attitudes towards enrichment?
Line 107 – Perhaps you ‘natural’ lifestyle instead of ‘original’?
Line 109 – 112 – This sentence needs revising. ‘Because they felt’ or alternatively dropping the ‘because’ an leaving ‘who felt’ on its own would be correct.
Line 112 – What longstanding practices? Some discussion of exactly what practices caregivers felt were being questioned would be useful to add context, as would some introduction of why mimicking wild-type behaviours in the zoo context was considered an uncomfortable subject. If the reasons are unclear or not well-studied this would be good to mention here as a segue into the survey.
Line 119 – 120 – These paragraphs could be merged.
Line 155 – Can the authors clarify the purpose of a ‘considering implementation’ response? Of course as this study was conducted 20 years ago retroactively changing this is not possible, but some explanation may be beneficial. Considering implementation could imply that there is still a barrier, but it could also imply that there isn’t a barrier and they just aren’t there yet, which are two different responses.
Line 164 – ‘Foraging devices’ would be clearer.
Line 168 – 170 – Explicit clarification that these are not strictly environmental enrichment practices would be beneficial. Though they could be, visitors touching the animals for example is not broadly appropriate and environmentally conscious exhibits could mean any number of things that do not necessarily include naturalistic enrichment practices.
Line 173 – Please include the reference for this report.
Line 211 – Perhaps you would consider using the late Valerie Hare.
Line 251 – 252 – These paragraphs could be merged.
Line 316 – 317 – Can you provide more specific examples, and how they substantially differ from the prevision iteration of the law? Are there any gaps the new legislation do not address? More evidence that the perspective in Japan is changing would help to strengthen that point. For example, while there may be more restrictions, private ownership of wild non-domestic species in many localities continues to rise. Does this impact zoos and their missions in any way?
Line 343 – Are there any references to support this? There are many papers published from Japanese zoos covering many topics, including training and environmental enrichment, that could exemplify the point that animal wellbeing approaches have shifted and changed since the survey was conducted.
Comments on the Quality of English LanguageMinor editing of English language required
Author Response
Yamazaki and I sincerely appreciate your valuable comments and suggestions. We are strongly encouraged by the positive feedback we received from you, reflected in the following in the manuscript.
Regarding the lack of references: Thank you very much for your essential references. To delve into the diverse cultural perspectives on animal welfare and to provide a clearer vision of the future direction of animal welfare in zoos, Yamazaki and I have created a new section, "5.2. Future Discussion," to provide the necessary context for further discussion. We believe this makes the conclusions and recommendations of this paper a more scholarly contribution.
Regarding further analysis of the survey results: Yamazaki and I stated in the notes that we cannot comprehensively analyze the demographics of the surveyed zoo staff because they are beyond the limits of the available data set. Now, we think 20 years after the survey, it is necessary to do a follow-up study. At that time, we plan to conduct a more comprehensive statistical analysis, including differences in opinion by gender and career stage.
Below are the revisions for each section. Thank you so much for all your feedback.
Line 9, Line 30
The word "Misinterpreting" was corrected to lowercase.
Line 44 - 45
Paragraphs were merged to improve the consistency of content.
Line 52, 58 - 59
We merged "animal well-being" into "animal welfare." This improves the clarity of the paper.
Line 66 - 71
Case studies from Thailand, Korea, and the Philippines are now presented in detail. This makes introducing the acceptance of animal welfare principles in non-Western countries more evident. However, your suggestion to refer to case studies from the Western zoo community and specifically compare the state of acceptance of environmental enrichment and animal welfare in Western and non-Western Europe, including a detailed discussion of the cultural and historical framework and spirituality, is beyond the capacity of Yamazaki and myself, and therefore, is paper could not be done. Nevertheless, this comparison and discussion are necessary in the future to help readers understand this subject from a broader perspective. Thank you for your suggestions.
Line 74
Sorry, we could not determine how we should add a discussion on the 3Rs about scientific standards. We keep your suggestion in mind for future issues.
Line 99 - 101
We added details about papers that have influenced attitudes toward environmental enrichment.
Line 107
Changed "original" to "natural."
Line 109 - 112
Corrected sentences and improved clarity.
Line 112
We added specifics about long-term practice. This has given context to the section and improved understanding.
Line 119 - 120
Paragraphs have been merged to improve the consistency of content.
Line 155
We added an explanation of the nuance of the response "considering implementation.
Line 164
Foraging targets" was changed to "Foraging devices.
Line 168 - 170
It is clarified that these efforts are not environmental enrichment practices to avoid misunderstanding content.
Line 173
Added references. This clarifies the sources of information in the section.
Line 211
Reworded to "late Valerie Hare." Our deepest condolences.
Line 251 - 252
Merged paragraphs to improve the consistency of content.
Line 316 - 317
We added a discussion of changes in Japanese views due to changes in the law.
Line 343
We added references.
Reviewer 4 Report
Comments and Suggestions for Authors
The paper submitted for review on the topic: "Bewilderment in the early stage of the introduction of environmental enrichment in Japanese zoos" is interesting because it deals with a sensitive topic about the welfare of animals in zoos. The scientific work has a serious drawback when it comes to data processing. In the presence of 1243 studied members, a much more serious analysis of variance should be done for the reasons why they gave the following results. In this line of thought, I believe that the work should be fundamentally reworked, which will probably yield several reasons for the results. As a result, I assume that the conclusion will also take on a completely new look.
Author Response
Yamazaki and I express our deepest gratitude for your valuable feedback. We apologize for the delay in responding to your review. As you pointed out, I also fully understand the importance of data processing highlighted in your comments, but I have a practical limitation. Yamazaki and I did not directly collect the dataset used in this paper. It is based on a survey conducted by the Japan Association of Zoos and Aquariums (JAZA) in 2001, and Yamazaki and I cannot access the complete data set. Therefore, regretfully, Yamazaki and I cannot perform a variance analysis on this dataset.
Yamazaki and I focused on extracting as much insight as possible from the existing data. Specifically, focusing on zookeepers' bewilderment towards the introduction of animal welfare and environmental enrichment in Japanese zoos in the early 2000s, we draw on national survey data and related public documents to find out whether the acceptance of animal welfare in Japanese zoos is hindered and carefully reasoned that this bewilderment is due to a combination of factors. (e.g., environmental enrichment as an exhibition strategy, lack of information on animal welfare, the popularity of mobile zoos in Japan after World War II, and Japanese animal protection that emphasizes longevity over the emotions of captive animals law)
However, as you rightly pointed out, a more extensive dataset and variance analysis would add further depth to the research. In the future, Yamazaki and I aim to conduct more extensive data collection for Japanese zoo staff and apply statistical methods to obtain a more comprehensive analysis of the current situation and 20 years of change.
Round 2
Reviewer 2 Report
Comments and Suggestions for Authors
The main problems detected in my first review remain unchanged: the discussion relies heavily on unpublished data from a survey carried out in 2001. It discusses the results of the survey without showing methodological details. The questionnaire is not included, only the analysed data. Importantly, the research involving humans was not appreciated by an ethics committee or exempted by legislation. Therefore in my opinion the paper cannot be accepted for publication.
Author Response
Thank you so much again for your valuable comments.
Ethics Committee approval of the study: As you have indicated, research involving human subjects requires Ethics Committee approval. Authors are unable to ascertain what ethical approval process it went through, because this survey conducted by the Japanese Association of Zoos and Aquariums in 2001. However, given that it is part of a regular survey conducted with the approval of JAZA, that the responses were made with the voluntary consent of the participants, and that the information collected in the survey was processed in a manner that does not identify individuals, we certain that the ethical validity of the study meets current ethical standards.
Reviewer 4 Report
Comments and Suggestions for Authors
I recommend that the authors, in the next study, carefully describe the research methods.
To take into account all the factors that can have an influence on the answers and attitudes of the people being studied.
Author Response
We would like to thank you again for your valuable suggestions. Yes, we would like to get a dataset to consider detailed influencing factors in our next study.